# Exploring Variability in Landscape Ecological Risk and Quantifying Its Driving Factors in the Amu Darya Delta

**DOI:** 10.3390/ijerph17010079

**Published:** 2019-12-20

**Authors:** Tao Yu, Anming Bao, Wenqiang Xu, Hao Guo, Liangliang Jiang, Guoxiong Zheng, Ye Yuan, Vincent NZABARINDA

**Affiliations:** 1State Key Laboratory of Desert and Oasis Ecology, Xinjiang Institute of Ecology and Geography, Chinese Academy of Sciences, Urumqi 830011, China; yutaogis@gmail.com (T.Y.); xuwq@ms.xjb.ac.cn (W.X.); liangliang.jiang@ugent.be (L.J.); zhengguoxiong17@mails.ucas.edu.cn (G.Z.); yeyuanrs@gmail.com (Y.Y.); vincentnzabarinda@yahoo.com (V.N.); 2Key Laboratory of GIS & RS Application Xinjiang Uygur Autonomous Region, Urumqi 830011, China; 3University of Chinese Academy of Sciences, Beijing 100049, China; 4Research Center for Ecology and Environment of Central Asia, Chinese Academy of Sciences, Urumqi 830011, China; 5School of Geography and Tourism, QuFu Normal University, Rizhao 276825, China; guohao@qfnu.edu.cn; 6Sino-Belgian Joint Laboratory of Geo-information, 9000 Ghent, Belgium

**Keywords:** landscape ecological risk, biophysical and socioeconomic driving factors, geographically weighted regression, Amu Darya Delta

## Abstract

Examining the drivers of landscape ecological risk can provide scientific information for planning and landscape optimization. The landscapes of the Amu Darya Delta (ADD) have recently undergone great changes, leading to increases in landscape ecological risks. However, the relationships between landscape ecological risk and its driving factors are poorly understood. In this study, the ADD was selected to construct landscape ecological risk index (ERI) values for 2000 and 2015. Based on a geographically weighted regression (GWR) model, the relationship between each of the normalized difference vegetation index (NDVI), land surface temperature (LST), digital elevation model (DEM), crop yield, population density (POP), and road density and the spatiotemporal variation in ERI were explored. The results showed that the ERI decreased from the periphery of the ADD to the centre and that high-risk areas were distributed in the ADD’s downstream region, with the total area of high-risk areas increasing by 86.55% from 2000 to 2015. The ERI was spatially correlated with Moran’s I in 2000 and 2015, with correlation of 0.67 and 0.72, respectively. The GWR model indicated that in most ADD areas, the NDVI had a negative impact on the ERI, whereas LST and DEM had positive impacts on the ERI. Crop yield, road density and POP were positively correlated with the ERI in the central region of the ADD, at road nodes and in densely populated urban areas, respectively. Based on the findings of this study, we suggest that the ecological constraints of the aforementioned factors should be considered in the process of delta development and protection.

## 1. Introduction

A stable ecosystem is the basis for the harmonious development of nature and society. However, with the intensification of human activities and the changing environment, natural ecosystems have been disturbed by urban expansion, population growth and natural disasters (e.g., floods [1] and debris flow [2]), which have led to many ecological risks [3,4]. Ecological risk reflects the possibility of degradation of an ecosystem subjected to external pressures [5,6]. Higher ecological risks may increase the likelihood of ecological problems, such as biodiversity loss, habitat fragmentation and environmental pollution [7,8]. As an important branch of ecological risk, landscape ecological risk refers to the adverse effects of the interaction between landscape pattern and ecological process under the influence of natural or human factors [5]. Landscape ecological risk emphasizes the spatial scale effect of ecological risks and provides a way to spatially represent multi-source ecological risks [9]. Therefore, landscape ecological risk assessment can provide a theoretical basis and technical support for regional landscape ecological construction [10].

In recent years, many studies have investigated the landscape ecological risk in various regions, [5,8,11], and the driving factors of landscape ecological risk have attracted much attention worldwide. Mo et al. (2017) focused on the relationships between road network expansion and node effects and regional landscape ecological risk and found that roads can affect the degree and spatial pattern of regional landscape ecological risk [12]. Yuan et al. (2019) studied the changes in landscape ecological risk under the background of urbanization in the Qinhuai River basin and reported that with increasing urbanization level, landscape ecological risk increased [1]. Some studies have addressed the spatial changes in landscape ecological risk based on land use change [6,11]. All of the aforementioned studies have contributed to a better understanding of the impacts of human activities on landscape ecological risk. However these studies have mainly focused on a single factor when attempting to analyse quantitatively the relationships between landscape ecological risk and driving factors [9,12]. The degree and spatial pattern of landscape ecological risk are affected by multiple factors, such as nature, society and the economy [7,13,14]. In addition, studies have shown that topography [15,16], changes in the surface environment [17,18], population [19,20] and crop yield [21] can affect landscape structure and functions of ecosystems [14], which in turn affect the degree and spatial distribution of landscape ecological risk. Therefore, focusing on the relationship between individual driving factors and landscape ecological risk may prevent the identification of factors that have critical impacts on the landscape ecological risk of a regional landscape and lead to incomplete scientific information for regional ecological planning [22,23]. It is necessary to explore the impacts of multiple potential social and biogeographic factors on regional landscape ecological risks and obtain a comprehensive understanding of the mechanisms underlying ecosystem risk variability.

Regression models have been adopted to describe quantitatively the relationships between driving factors and ecosystem changes [24,25]. For example, a linear regression model was used to study the impact of biophysical and socioeconomic factors on landscape changes [26]. A multinomial logistic model was selected to detect the driving factors of forest cover changes [27] and explore the forces underlying long-term urban expansion [28]. A maximum covariance analysis (MCA) was applied to reveal the relationship between the extension of a road network and landscape ecological risk [12]. However, spatial data (e.g., landscape ecological risk) may exhibit spatial autocorrelation and non-stationarity [9], which makes it difficult to meet the assumptions and requirements of conventional linear regression techniques (e.g., ordinary least squares, OLS). These regression methods can produce only “average” and “global” parameter estimates [29,30]; thus, they cannot handle spatial autocorrelation that exists in variables [9,31]. In recent years, a novel technology called geographically weighted regression (GWR) has been developed [32], which is a simple but effective method for exploring spatially varying relationships [9]. A GWR model allows for different relationships at different locations in a study area. Therefore, local parameters can be estimated instead of global parameters, which can better describe the relationships between local variables [33]. The GWR model has been widely used to explain changes in regional vegetation and urban surface environments or urban expansion [29,31,34,35,36]. Therefore, in this study, the GWR model was selected to study the relationships between changes in landscape ecological risk and the driving factors.

The Aral Sea was once the fourth largest inland lake in the world, providing rich ecosystem services and biodiversity for the region [37]. Moreover, the Aral Sea basin (ASB) is also a very important food production base in the world, and a large amount of cotton is produced in this area every year [38]. Representing one of the most shocking environmental disasters in the world, the area of the Aral Sea has greatly declined due to multiple factors, such as the continued expansion of agriculture, inappropriate irrigation and frequent land reclamation [39]. The ecosystem around the Aral Sea has been largely destroyed, particularly in the Amu Darya Delta (ADD) [40]. Characterized by a fragile ecological environment, the ADD has undergone soil salinization [41] and experienced sandstorms and climate change [42]. Due to both natural factors and human activities, such as extensive expansion of farmland and construction land [43], grassland and forestland in this area have been extensively degraded [44,45], and the structure and function of the landscape are being damaged. These phenomena have led to high landscape ecological risk. It is important to examine the driving factors of landscape ecological to provide a scientific reference for upcoming planning and landscape optimization. The main objectives of this study were to (1) reveal the spatial variability of landscape ecological risk in the ADD over the past 15 years and (2) use the GWR model to quantitatively distinguish the relationships between landscape ecological risk and selected potential driving factors. Our research provides a reference for future assessments of landscape ecological risk and ecological protection in other delta areas.

## 2. Materials and Methods

### 2.1. Study Area

The ADD is located downstream of the Amu Darya and crosses Uzbekistan and Turkmenistan (Figure 1), with an area of 6.3 × 10^4^ km^2^. The Amu Darya water sources are large permanent glaciers and snow-covered areas. The Amu Darya is the main source of water in the Amu Darya basin [46], which provides irrigation water for farmland and domestic water. As the ADD takes in water at the end of the Amu Darya, it is severely affected by large-scale changes in hydrological conditions [47]. The ecological environment of the ADD is sensitive to changes in natural conditions and human activities.

The ADD has an extremely continental climate characterized by extremely dry conditions throughout the year, with hot summers and cold winters. The annual average temperature is approximately 13 degrees Celsius, and the average frost-free period is 205 days [48]. The potential annual evapotranspiration is 1400–1600 mm, and the average annual precipitation is only 100 mm [43]. The dry climate makes the ADD one of the most sensitive and ecologically fragile regions in the world. The initial landscape types in the delta were primarily marsh wetlands, reed forests and typical riverine forests [43]. However, in recent decades, this region has experienced rapid economic and population growth with the conversion of forests and grasslands to agricultural and construction land, and the fragile ecological environments are highly sensitive to changes in natural factors [44,49]. These socioeconomic and biogeographic factors shape the ecological processes in the region and contribute to the increasing ecological crisis and risks. To mitigate the gradual deterioration of the ecological environment of the ADD, ecological restoration projects have been implemented [50]. However, the impacts of biogeographic and socioeconomic driving factors on the landscape ecological risks of this region remain unclear. Understanding the ecological structure and function of the ADD and exploring the impacts of these potential drivers on the ecological environment are important.

### 2.2. Data Description and Processing

#### 2.2.1. Land Use/Land Cover (LULC)

The land-use/land-cover (LULC) dataset for 2000 and 2015 in ArcGIS shapefile format used in this study was developed by the Chinese Academy of Sciences through the Database of Global Change Parameters, with the data interpreted based on Landsat images (http://www.eercasia.com/# [51]). According to previous studies [11,12,15] and the environmental characteristics of the ADD, the land use types were divided into seven categories: farmland, forest land, grassland, construction land, water, wetland, and unused land.

#### 2.2.2. Biogeographic Variables

According to previous studies and data availability, digital elevation model (DEM) [15], land surface temperature (LST) [35,52] and normalized difference vegetation index (NDVI) [3] data were selected as the biogeographic factors. DEM data were obtained from NASA with a spatial resolution of 90 m (https://search.earthdata.nasa.gov/search). LST data were derived from the 8-day composite MOD11A2 products at 1 km spatial resolution (https://search.earthdata.nasa.gov). The LST data were further aggregated into monthly and annual averages for 2000 and 2015. We used the NDVI data from the MOD13A1 product, which originated from the National Oceanic and Atmospheric Administration (NOAA)/Advanced Very High-Resolution Radiometer (AVHRR) land dataset at a spatial resolution of 1 km and a temporal resolution of 16 days for the study periods (https://search.earthdata.nasa.gov). We chose the maximum value composite (MVC) method to obtain the NDVI values for the year 2000 and 2015. The MVC method is a procedure used in satellite imaging, which is applied to vegetation studies [53,54,55,56]. It requires that a series of multitemporal geo-referenced satellite data be processed into NDVI images. On a pixel-by-pixel basis, each NDVI value is examined, and only the highest value is retained for each pixel location [56].

#### 2.2.3. Socioeconomic Variables

Collecting detailed and complete socioeconomic data in this region is difficult. Hence, we selected three types of data that could be obtained for free as socioeconomic drivers, population density (POP), crop yield (tons/ha) and road network data, to investigate their relationships with landscape ecological risk. Because agricultural production remains the principal socioeconomic activity of the ADD [51,57], we selected the 2000 and 2015 crop yield data provided by the WUEMoCA database (http://wuemoca.net/app/#) as one of the socioeconomic drivers in this study. In recent decades, the population of the ADD has experienced large-scale growth, which has resulted in a series of negative impacts to the ecological environment [57]. POP data for 2000 and 2015 were downloaded from WorldPop (https://www.worldpop.org/geodata) with a spatial resolution of 1 km. Road network will affect the ecological environment while promoting regional economic development [7,58]. Previous research has shown that road networks as part of socio-economic factors can exacerbate the shift in landscape types [59,60,61], particularly driving the conversion of forest land to non-forest land [62]. Kernel density estimation has been proven to better reflect road density and network characteristics [63,64], and are linked to landscape ecological risk [9,12]. Therefore, we used the methods used in previous studies [12] to obtain the road kernel density (KD) data, and the road data were extracted in 2015 from OpenStreetMap (https://www.openstreetmap.org [65]).

## 3. Methods

### 3.1. Construction of the Ecological Risk Index (ERI)

#### 3.1.1. Landscape Disturbance Index (LDI) 

The landscape disturbance index (LDI) correctly reflects the fact that external disturbances [12], including biogeographic and socioeconomic factors, positively affect ecosystems represented by different landscapes, which include a specific combination of the fragmentation index (*F_i_*) (indicating the degree of patches fragmentation for a certain landscape type [9,66]), splitting index (*S_i_*) (indicating the degree of patches separation for a certain landscape type [9,13]) and dominant index (*D_i_*) (describing the degree of patches importance for a certain landscape type [9,66]). The formulas for these indexes are as follows:(1)$$F_{i}=\frac{n_{i}}{A_{i}}$$
(2)$$S_{i}=\frac{A}{2A_{i}}\sqrt{\frac{n_{i}}{A}}$$
(3)$$D_{i}=\frac{\left(Q_{i}+M_{i}\right)}{4}+\frac{L_{i}}{2},Q_{i}=\frac{n_{i}}{N},M_{i}=\frac{B_{i}}{B},L_{i}=\frac{A_{i}}{A}$$
(4)$$LDI_{i}=aF_{i}+bS_{i}+cD_{i}$$
where *n_i_* is the number of patches of landscape *i*; *N* is the total patch number; *A_i_* is the area of landscape *i*; *A* is the total area; *Q_i_* is the ratio of the number of patches of landscape *i* to the total patch number; *M_i_* is the ratio of samples for landscape *i* to the total number of samples; *B_i_* is the number of samples for landscape *i*; *B* is the total number of samples; *L_i_* is the ratio of the area of patches of landscape *i* to total area; and *a*, *b* and *c* represent the weights of the landscape metrics, where *a* + *b* + *c* = 1; according to previous studies [6,9,12], these were set as *a* = 0.5, *b* = 0.3, *c* = 0.2.

#### 3.1.2. Landscape Fragility Index (LFI)

The landscape fragility index (LFI) represents the vulnerability of the internal structure of the ecosystem. This index mainly measures the stability or anti-interference ability of the landscape structure itself [12]. A higher LFI indicates lower resistance and more significant ecological risk. Previous studies have provided methods for determining the vulnerability of different landscape types [11,12]. In this study, the vulnerability of the seven landscape types, consisting of construction land, forest land, grassland, farmland, water, wetland, and unused land were ordered from lowest to highest. Vulnerability was normalized to generate a vulnerability index for the corresponding landscape types.

#### 3.1.3. Ecological Risk Index (ERI)

Based on the LDI and LFI mentioned above, we constructed the landscape ecological risk index (ERI). This index can accurately describe the relative sizes of integrated ecological losses in a specifically selected sample and can adequately reflect the possible changes in ecological risks attributed to landscape pattern changes. Furthermore, the indicator can be used to transform the landscape structure into spatial ecological risk variables through sampling [12]. The formula that was used is as follows [9,11,13]:(5)$$ERI=\sum_{i}^{j}\frac{A_{ki}}{A_{k}}\sqrt{LDI_{i}\cdot LFI_{i}}$$
where *ERI* represents the landscape ecological risk index; *A_ki_* is the area of landscape *i* in sample *k*; *A_k_* is the area of sample k; and *j* is the number of landscape categories in sample *k*.

Referring to previous studies and the characteristics of landscape coverage in the study area, we used the equal interval method to divide the study into a total of 1606 sample units with a grid cell size of 5 × 5 km for spatial analysis of the ERI (Figure 2). Then, the ERI was calculated for each sample area as the ecological risk value at the centre of the sample area. The ERI of the entire study area was obtained by interpolating the total of 1606 cell points via ordinary kriging in ArcGIS 10.2 software [12,15]. Finally, the natural breaks method was used to divide the ERI into five levels: low risk, sub-low risk, medium risk, sub-high risk and high risk [9,11].

### 3.2. Spatial Autocorrelation Analyses

Spatial autocorrelation analysis is a commonly used method of spatial statistics and can quantitatively describe the spatial distribution characteristics of statistics [67]. Xue et al. (2019) and Hu et al. (2018) revealed the distribution patterns of regional landscape ecological risk and ecological security, respectively, by using spatial autocorrelation analysis [15,68]. In this study, the global Moran’s I index and local indicators of spatial association (LISA) were employed to analyse the spatial statistics of the ERI.

The global Moran’s I provides an overall measure of spatial autocorrelation, with Moran’s I ranging from +1 to −1 [69]. Moran’s I values >0 would indicate positive spatial autocorrelation of the ERI; values closer to +1, indicate strong spatial autocorrelation of the ERI. Moran’s I = 0 indicates no spatial autocorrelation, and the ERI exhibits a random spatial distribution; Moran’s I values <0 would indicate negative spatial autocorrelation of the ERI. LISA represents a set of local statistical methods that can measure the spatial autocorrelation of each observed object in a spatially adjacent area to reveal the spatial clustering pattern of the research object. The spatial clustering can be defined using the local Moran’s I index [67]. A positive LISA indicates that the object value is similar to the neighbouring values and exhibits a high-high cluster (H-H, high value in a high-value neighbourhood) or low-low cluster (L-L, low-value in a low-value neighbourhood) spatial clustering mode. A negative LISA indicates potential spatial outliers of target values, including high and low outliers (H-L, high values in low-value neighbourhoods) and low-high outliers (L-H, low values in high-value neighbourhoods), that are significantly different from the values in the surrounding area [67].

### 3.3. Geographically Weighted Regression (GWR) Model

As a form of regression model, the GWR has demonstrated superiority in reflecting the spatial variation in observed variables by considering the spatial position of the sample [70]. This regression method is different from a traditional linear regression because it allows for local parameter estimation. The results of these parameters show how the relationships between variables changes throughout the space, and local spatial patterns can be detected and estimated to understand the possible causes of this pattern [71]. In GWR, the parameters for each observation at each location can be estimated by weighting all observations around a specific point according to their spatial proximity. Gaussian distance decay can be used to express the weighting function:(6)$$Wij=exp\left(\frac{d_{ij}^{2}}{h^{2}}\right)$$
where *Wij* represents the weight of observation *j* for location *i*, *d_ij_* is the Euclidean distance between points *i* and *j*, *h* is a kernel bandwidth that affects the distance-decay of the weighting function. The kernel bandwidth [31,72] is very important when building a GWR model, which is usually determined by cross-validation (CV) or the Akaike information criterion (AICc) [72,73]. Since the AICc method considers the difference in the degree of freedom of different models compared to the CV method, it can solve the problem more quickly and conveniently. In this paper, we use a Gaussian function to determine the weight and the AICc method to determine the optimal bandwidth [9,64]. The GWR equation is defined as follows:(7)$$y_{i}(u_{i},v_{i})=\beta_{o}(u_{i},v_{i})+\sum \beta_{j}(u_{i},v_{i})x_{i}+e_{i}(u_{i},v_{i})$$
where *y_i_* is the value of the ERI for the *i* th grid; *x_i_* is the selected factor of ERI (e.g., DEM, LST, NDVI, POP, crop yields, road KD); *β_o_* and *β_j_* are the coefficients of the constant and the explanatory variables, and *e_i_* is the stochastic error term; (*u_i_*, *v_i_*) is the geographic coordinates of the *i*th grid. The GWR model generates a coefficient for each grid, and the coefficient is obtained by the following formula:(8)$$\hat{\beta }(u_{i},v_{i})={\left(X^{\prime }w(u_{i},v_{i})X\right)}^{-1}X^{\prime }w(u_{i},v_{i})Y$$
where *w*(*u_i_*, *v_i_*) is the spatial weight matrix, which is unique for each grid [64]. In this study, the biogeographic and socioeconomic variables were resampled to 5 × 5 km grids by using the bilinear interpolation method [45,74,75] to match the ERI data and used the spatial modelling tool in ArcGIS 10.2 software to calculate the GWR parameters.

## 4. Results

### 4.1. Landscapes Change

During the study period, the dominant landscape in the study area remained farmland, which accounted for almost 56% and 58% of the total area in 2000 and 2015, respectively (Figure 3). Unused land was also widely distributed in the ADD, mainly in the lower reaches and the edge of the ADD, occupying 17.07% and 19.93% of the total area in 2000 and 2015, respectively. Grassland accounted for 15.82% and 10.65% of the total area in 2000 and 2015, respectively. Forest lands, water and wetlands were the least extensive landscape types in the ADD, each accounting for less than 7% of the total area. Forest land was mainly distributed along the Amu Darya.

Figure 4 shows how the landscapes have changed over the 15 years. Over the period 2000–2015, there were increases in the proportions of farmland, unused land and construction land in the ADD. Construction land and unused land increased by 16.83% and 16.79%, respectively, while farmland increased by 5% over the past 15 years. However, opposite trends were observed for wetland, water, grassland and forest land areas, which showed varying degrees of decline. Among these, the most obvious reductions were in forestland and grassland, which decreased by 37.55% and 32.70%, respectively.

### 4.2. Changes in Landscape Metrics

As shown in Figure 5, there were pronounced changes in the landscape indexes from 2000 to 2015. In terms of fragmentation (Figure 5a), as measured by the landscape fragmentation index, the wetland and water areas possessed high values in 2000 and 2015, whereas the farmland, construction land and unused land had low values in both years. The results revealed that among the landscape types wetland and water landscapes exhibited the highest degrees of patch separation. Additionally, farmland, construction land and unused land exhibited low degrees of patch separation. The extent of landscape fragmentation in 2015 was higher than that in 2000 for all landscape types, which implied that landscape fragmentation had increased during the study period. The landscape fragmentation index of forest land and grassland increased significantly between the two years, revealing rapid increases in the fragmentation of forest land and grassland patches.

In terms of dominance (Figure 5b), the landscape dominance index values for farmland and unused land were higher than those for the other landscape types, indicating that farmland and unused land were the dominant landscape types in the ADD according to the size of the area of each landscape (Figure 3). The landscape dominance index values of the farmland and unused land showed increasing trends from 2000 to 2015, whereas those values of forest land, grassland, water and wetland decreased over the studied period. The landscape dominance index of construction land changed little over the studied period.

In terms of separation, wetlands had the highest landscape splitting index value among the landscape types, followed by water (Figure 5c). During the studied period, the separation of all the landscape types showed an increasing trend, with that of forest land increasing significantly over the 15 years. However, the changes in landscape splitting index for farmland, construction land, unused land and grassland were small, with weakly increasing trends from 2000 to 2015.

### 4.3. Changes in Ecological Risk Index (ERI)

Figure 6 shows the spatial distribution of the ERI in the ADD in 2000 and 2015. The figure reveals that the ERI distribution varied over space and time. In 2000 (Figure 6a), the high-risk areas were mainly located at the periphery of the ADD, with some small zones near the city of Nukus, whereas the sub-high-risk areas were mainly distributed in the lower reaches of the ADD. Furthermore, most of the low-and sub-low risk regions were located in the west and central region of the ADD, and the medium-risk zones were mainly distributed at the periphery of the ADD. The most significant changes in the spatial distribution of the ERI from 2000 to 2015 (Figure 6b) were in the high-risk regions, which were mainly located in the lower reaches of the ADD. Moreover, the spatial distributions of the ERI (Figure 6) in 2000 and 2015 showed that the risk degree gradually decreased from the outside to the inside of the study area, indicating that the degree of risk on the outside of the study region was relatively higher than that on the inside of the study region during the research period.

The areas and variations in different levels of the ERI during the study period are shown in Table 1. The ecological risk levels of low and sub-low risk accounted for more than half the total area of the ADD in both 2000 and 2015. The high ecological risk zones covered 2832.35 km^2^ and 5283.65 km^2^ in 2000 and 2015, respectively, which indicated that the total area of high-risk areas exhibited a significant increase trend (by 86.55%) from 2000 to 2015. The total area of medium ecological risk areas decreased from 6718.05 km^2^ in 2000 to 6186.15 km^2^ in 2015, representing a decrease of 7.92%, and the sub-high ecological risk areas represented 17.86% and 12.8% of the study area in 2000 and 2015, respectively.

Using intersect tool, as implemented in ArcGIS, we created a spatial map of risk levels changes in the ADD between 2000 and 2015, and there were 21 different combinations. The combinations with no risk level transformation or smaller conversion areas were merged into “Stable or little change”. Finally, 10 combinations of risk level transformation were formed as shown in Figure 7. Among them, there were five combinations that transformed from low level risk to high level risk and four combinations that transformed from high level risk to low level risk. As shown in the figure, most risk levels remained stable or little changes spatially during the study period, although large changes were observed in the lower reaches of the ADD, mainly comprising changes from sub-high to high ecological risk. Changes from the sub-low to low ecological risk levels mainly occurred in the Turkmenistan (TKM) region of the ADD. The changes in risk level showed that the regions of higher risk degree were mainly in the Uzbekistan (UZB) part of the delta, especially in the downstream part of the ADD, whereas the regions with lower risk degree were concentrated in TKM. Table 2, shows the changes in the distribution of risk levels in the TKM and UZB regions of the ADD. High ecological risk areas increased by almost 160%, but other risk levels all showed a downward trend in the UZB part of the ADD over the past 15 years. In the TKM, the low ecological risk increased by almost 20% and other ecological risk levels tended to decrease, with the high ecological risk decreasing significantly (54%).

### 4.4. Spatial Autocorrelation of ERI

GeoDa is a free and open source software tool that serves as an introduction to spatial data analysis. Using GeoDa 1.14 (University of Chicago, Chicago, IL, USA) (http://geodacenter.github.io/download.html), we conducted spatial autocorrelation analysis of the ERI values in 2000 and 2015.The values of Moran’s I were 0.669318 and 0.718535 in 2000 (Figure 8a) and 2015 (Figure 8b), respectively. The global Moran’s I value of the ERI was greater than zero during the studied period, which revealed spatial clustering of the ERI, which also indicates a strong positive correlation. Moran’s I of the ERI significantly increased from 2000 to 2015, indicating that the spatial clustering in ERI distribution was enhanced over time in the ADD.

A spatial LISA cluster map for the ERI in 2000 and 2015 was obtained (Figure 8c,d). The spatial distribution of the ERI in the study periods was dominated by the “L-L” and “H-H” trends. The “L-L” areas were mainly distributed in the central ADD, and the “H-H” areas were mainly concentrated on the edge and lower reaches of the ADD in 2000 and 2015. However, the “H-H” areas that were distributed in the lower reaches of the ADD exhibited a significant increasing trend from 2000 to 2015, which revealed that the areas with high ecological risk in this region increased significantly.

### 4.5. Geographically Weighted Regression (GWR) Model

#### 4.5.1. Biogeographic Factors

Using the ERI as the independent variable and biophysical and socioeconomic variables as explanatory factors, we mapped the correlation coefficients between the various variables and the ERI at the grid level through the GWR model (Figure 9 and Figure 10). In the GWR model, the coefficients described statistically significant associations between the independent variable and explanatory factors; coefficients >0 revealed a positive relationship while coefficients <0 revealed a negative relationship with the independent variable.

Figure 9a,d show the spatial distribution of the coefficients of the correlation between LST and the ERI in 2000 and 2015, respectively. A positive relationship between LST and ERI was observed in most areas of the delta, and a small portion of the northern delta region exhibited a negative relationship in the two years. The area with positive correlation coefficients (red colour) increased from 2000 to 2015 and became concentrated in the TKM region in 2015, and the negative-relationship areas increased in the north of the ADD.

In contrast to the patterns observed with the LST, a negative correlation between the NDVI and ERI was observed in most areas of the ADD, and a small part showed a positive correlation (Figure 9b,e). Over the 15 years, the areas with positive relationships tended to decrease in the ADD, although an upward trend was observed in the lower parts of the ADD. The coefficients for the DEM were mapped in Figure 9c,f, which showed that most regions showed a positive relationship of DEM with the ERI in 2000 and 2015 and that there were areas with negative relationships in the northern part of the ADD. From 2000 to 2015, the areas with strong positive correlations increased significantly, which were mainly located in the southern part of the ADD (the regions in red).

#### 4.5.2. Socioeconomic Factors

Figure 10a,d show the correlation coefficients between the POP and ERI at the grid level according to the GWR model. The positive correlation areas were mainly distributed in the centre of the ADD in 2000 and 2015. However, in the lower reaches of the ADD, the areas with positive correlations increased between 2000 and 2015. The areas with negative coefficients for POP were mainly located at the borders of the ADD during the study period.

Figure 10b,e show that the areas with positive coefficients for crop yield were located in the central region of the ADD and that from 2000 to 2015, the positive-coefficient areas were reduced near northern Nukus. Most areas with negative correlations were located in the north and south of the ADD, and areas with small negative coefficients increased in the upper reaches and lower reaches of the ADD over the 15 years.

The coefficients for road KD in 2000 and 2015 are mapped in Figure 10c,f, respectively. In 2000 and 2015, most of the ADD was represented by negative coefficients, and the smallest coefficients (the regions in blue) were mainly located within the boundaries of the ADD. The positive coefficients mainly occupied the nodes of the road network and areas near the cities during the study period.

## 5. Discussion 

### 5.1. Temporal and Spatial Patterns of the ERI

The spatial distribution of the ERI in 2000 and 2015 showed a consistent pattern, with higher ERI levels at the periphery of the ADD and lower ERI levels in the interior of the ADD (Figure 6). A possible reason for this pattern is that at the boundary of the ADD, the main landscape type was forest land; the fragmentation and separation of forest land was high (Figure 5), increasing the ERI at the ADD boundaries. On the other hand, this difference may be due to the poor soil and water conservation functions at the boundary of the ADD [57], the lack of management, and the inappropriate land use patterns, which led to more fragmented landscapes and increased the landscape risks [76]. The high landscape ecological risks at the edge of ADD serve as a reminder to policymakers to pay attention to the outlying regions of the regional ecosystems, such as delta margins. Because these regions are in the interlaced regions of the ecosystem [77], the ecological environment and ecosystem stability may be more sensitive to external disturbances [78], a possibility that should be considered when developing eco-planning policies.

The ERI inside the ADD is low, which may be related to the type of internal landscape. The interior of the ADD consists mainly of farmland and construction land. The fragmentation and separation of the landscape are small; moreover, the ecosystem in the internal delta is stable and strong and may produce lower ERI values, which is consistent with previous studies [5,12].

The greatest changes in the spatial distribution of the ERI were mainly concentrated downstream of the ADD. These changes correspond to the increase in the ERI from sub-high risk to high risk (Figure 7). The landscape ecological risk increased over time in the lower reaches of the ADD, which might be predominantly due to the degradation of a large number of grasslands to unused land (Figure 3 and Figure 4). Compared with the coverage in 2000, large areas of grassland have been degraded, resulting in increased grassland dispersal, landscape separation and fragmentation and ecological risks. In this study, we found that the ecology downstream of the ADD showed significant degradation, which has also been confirmed by previous studies [44,51,79].

The ERI of the UZB region of the ADD was significantly higher than that of the TKM region (Table 2). The UZB had a high level of ecological risk associated with large-scale degradation of grasslands and forestlands. In addition, studies have shown that due to agricultural expansion in the southwestern part of the ADD, a water supply channel has been established to supply water for agriculture in the TKM portion from the Amu Darya [51,80]. Therefore, most of the expanded farmland that was converted from highly fragmented grassland and forestland is concentrated in this area (Figure 3), which may improve the stability of the ecosystem and reduce the ecological risk of the area.

After the disintegration of the Soviet Union in 1991, UZB and TKM shifted from a socialist society to a capitalist society, and many individuals from rural populations migrated to the cities because of the transition from a planned economy to a free market [51,81]. The decline of the rural population and the subsequent disappearance of large agricultural subsidies have contributed to the abandonment of degraded farmland in the ADD, and this phenomenon is more common in UZB than in TKM [82,83]. In addition, the secondary vegetation that forms after the abandonment of farmland has a simple structure low biodiversity [15]. The abandonment of farmland not only entails an increased ecological risk for UZB but also has adverse socio-economic impacts, including increased food insecurity, a reduction in the number of households and increased unemployment [83].

In the last century, due to human activities, the amount of water transported from the Amu Darya into the Aral Sea decreased, triggering the Aral Sea crisis [37]. Maintaining a stable ecosystem of the ADD is extremely important for promoting the ecological environment of the Aral Sea and surrounding areas. Therefore, the landscape ecological risk of ADD should be addressed and efforts made to reduce risk to avoid further ecological disasters.

### 5.2. Effects of Biogeographic Factors on the ERI

In this paper, the impacts of biogeographic factors on the ecological environment in the ADD were quantitatively analysed using a GWR model, and the relationships between the landscape ERI and the selected driving factors of LST, NDVI and DEM were analysed. The LST can reveal substantial information about the ecological environment [84]. Changes in the LST can disturb the balance of matter and energy between the ground and atmospheric materials, resulting in changes in landscapes, climatic conditions [84,85] and vegetation [35], thereby affecting the local ecological environment. Furthermore, an increase in LST often reflects a deterioration of the regional ecological environment [35,86], as suggested by the results of this paper. In most areas of the ADD, the ERI and LST showed positive correlations (Figure 9a,d); the increases in LST will lead to increases in the regional landscape ecological risk.

However, in the northern part of the ADD, there were negative correlations in small regions. It is possible that soil moisture and humidity are sufficiently high in the northern part of the ADD [57] that the increase in surface temperature is not the main impacting factor in this region, which might explain the negative correlations. The NDVI is a good indicator of vegetation, which in turn reflects the state of the ecological environment [68]. The degradation of vegetation indicates the degradation of ecosystem function. Previous research has revealed pronounced vegetation degradation in the ADD [44]. In 2000 and 2015, strong negative correlations between NDVI and the ERI were observed in most areas (Figure 9b,e), indicating that vegetation has an inhibitory effect on landscape ecological risk of the ADD.

However, unexpectedly, some sporadic areas showed positive correlations, particularly in the lower delta. This result may be due to the fact that the landscape types in the lower parts of the Amu Darya are mainly forest land, grasslands and wetlands. The vegetation cover in this area is high and shows high NDVI values, although fragmentation of forest land, grassland and wetland landscapes occurs (Figure 5). A high degree of separation leads to increased landscape ecological risk; accordingly, a region with a positive correlation between the NDVI and ERI formed in the downstream regions of the ADD. The relationship between the DEM and the ERI exhibited a strong spatial pattern: in the middle and upper regions of the ADD, the DEM was positively correlated with the ERI, in the lower region, negative correlations were observed (Figure 9c,f), which is consistent with the results of previous research [15]. The DEM is one of the driving forces of changes in landscape type [87], and landscape ecological risks are closely related to landscape dynamics. The results of this study indicate that when using delta land, elevation should be considered when developing and using the upper and lower regions of the delta.

### 5.3. Effects of Socioeconomic Factors on the ERI

In 2000 and 2015, in the central part of the ADD, POP had significant positive correlations with the ERI (Figure 10a,d); suggesting that the increases in population size increased the landscape ecological risk in this area. These areas are mainly densely populated cities and their surrounding areas. Moreover, we observed a positive correlation between the downstream POP of the ADD and the ERI in 2015, which is inconsistent with the pattern observed in 2000. This result shows that the impact of human activities expanded from 2000 to 2015. The increase in POP has resulted in ecological risks for the sparsely populated areas in the lower part of the ADD. This result may be relevant to policymakers in the lower reaches of the ADD, where landscape ecological risks are particularly sensitive to population growth. At the periphery of the delta, POP is inversely related to the ERI, suggesting that population growth during the study period was not sufficient to influence the ecological risks of these areas. 

The ADD is the main food production base of the ASB, and has experienced a series of ecological and environmental problems due to inappropriate agricultural production activities [39]. As shown in this paper, the crop yield and the ERI are positively correlated in the major food production areas, such as the Nukus irrigation district (Figure 10b,e). The higher the yields of grain crops are in this region, the higher the degree of landscape ecological risk. Higher crop yields mean increases in the amount of land reclamation and agricultural irrigation and the degree of fragmentation of cultivated land, resulting in a higher ERI. 

In 2000 and 2015, the areas where the road network density was positively correlated with the ERI were mainly concentrated at the nodes of the city centres and the road network (Figure 10c,f), which is consistent with the results of previous studies [9,12]. However, negative correlations occurred at the ADD boundary, which confirms previous studies demonstrating that road networks are negatively correlated with the ERI in less dense areas [12].

## 6. Conclusions

Quantitative evaluations of the relationships between regional ecological risk and driving factors are of great significance for the prevention of regional ecological risk and landscape pattern optimization. To adequately explore the considerable influences of biophysical and socioeconomic factors on the landscape ecology of the ADD, we constructed the ERI of the ADD in Central Asia from 2000 to 2015 and used spatial autocorrelation and a GWR model to analyse the possible ERI changes and driving forces. The results showed that the distribution of the ERI exhibited a gradually decreasing trend from the periphery to the middle of the study area in both 2000 and 2015. The high-risk areas were mainly distributed in the downstream regions of the delta, and the total area of these areas increased by 86.55% from 2000 to 2015. The changes in the spatial distribution of the ERI revealed that most risk levels remained stable over the 15 years, although large transitions were observed in the lower reaches of the ADD, which were mainly caused by area transformations from sub-high ecological risk to high ecological risk.

The Moran’s I index values for 2000 and 2015 were 0.669318 and 0.718535, respectively, indicating pronounced spatial aggregation of the ERI. The risk “hot spot” areas were mainly distributed in the periphery and downstream regions of the delta, whereas the “cold spot” areas were mainly concentrated in the middle of the ADD.

The GWR model indicated that crop yield and the ERI exhibited significant positive correlations in the middle of the study area. Except for a few areas in the central region, the NDVI had a significant and negative impact on the ERI. Road density was positively correlated with the ERI in cities and roads, and in most areas of the ADD, the LST and DEM had positive impacts on the ERI. In 2000 and 2015, POP was significantly positively correlated with the ERI in the densely populated cities of the central delta and its surrounding areas. Moreover, the positive impact of population density on the ERI gradually extended from 2000 to the downstream regions of the ADD in 2015.

The ADD is adjacent to the Aral Sea, and its ecological environment has strong impacts on the stability and function of regional ecosystems. Land development and infrastructure construction should be reduced at the edges and downstream regions of the delta, and policies for the protection of forest lands and grasslands (such as grazing reduction policies) should be formulated to achieve ecological restoration. Furthermore, saline-alkali land control measures should be launched to increase land productivity, and drip irrigation technology should be encouraged through agricultural subsidies to alleviate the ecological pressure brought to the delta by agricultural production. Furthermore, ecological protection engineering can be considered to address the negative impacts of environmental variability on the delta ecosystem. 

The variations of landscape ecological risk are also impacted by national and local policies [15,88]. Due to the disintegration of the Soviet Union, the ADD showed changes in socio-economic patterns around 1991, the subsequent associated policies related to grazing, agricultural subsidies and water utilization have changed, leading to changes in vegetation cover and landscape types [89], which may affect the landscape ecological risk of ADD. In our research, we did not analyze changes in social institutions and local policies as the driving factor. Changes in regional landscape ecological risks before and after changes in social institutions and local policies should be studied in future research. Moreover, in order to further reveal the changes in landscape ecological risk and provide information for ADD landscape planning and ecological protection, the multiple time series spatio-temporal patterns change in landscape ecological risk based on the previous change-detection method [16] will be included in future research. 

## Figures and Tables

**Figure 1 ijerph-17-00079-f001:**
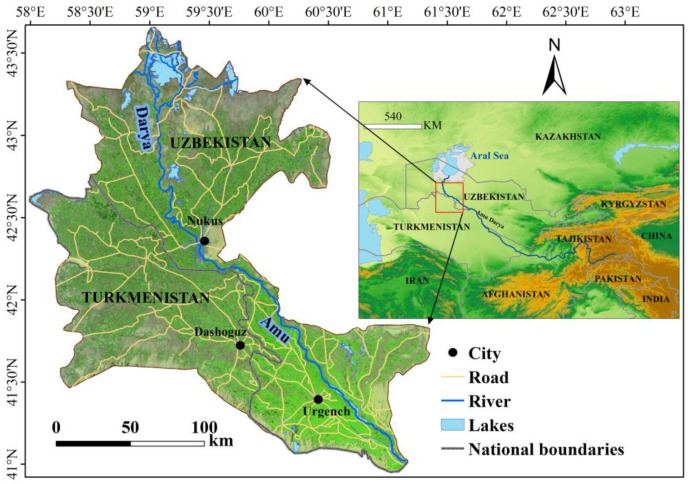
Location of the study area.

**Figure 2 ijerph-17-00079-f002:**
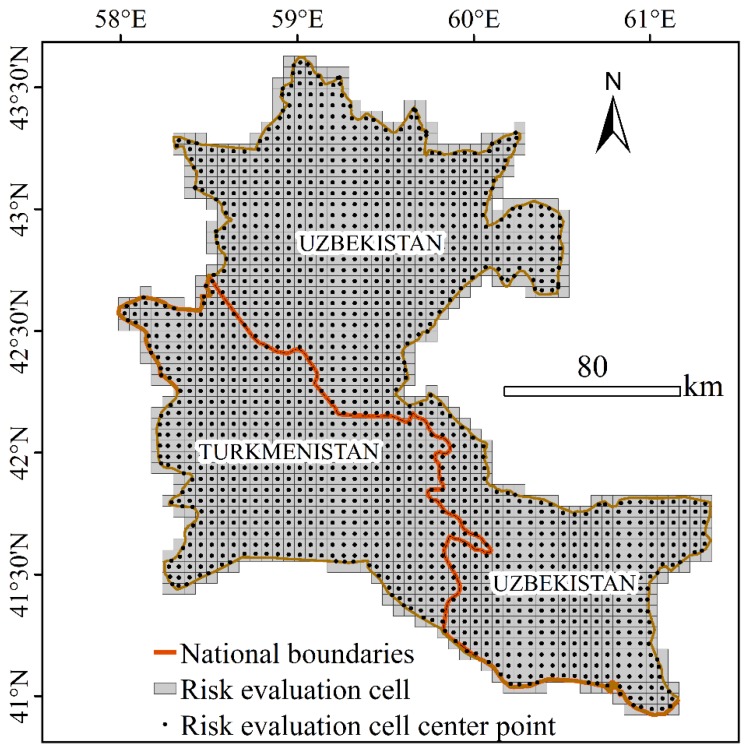
Ecological risk evaluation cells on a simple map of the Amu Darya Delta (ADD).

**Figure 3 ijerph-17-00079-f003:**
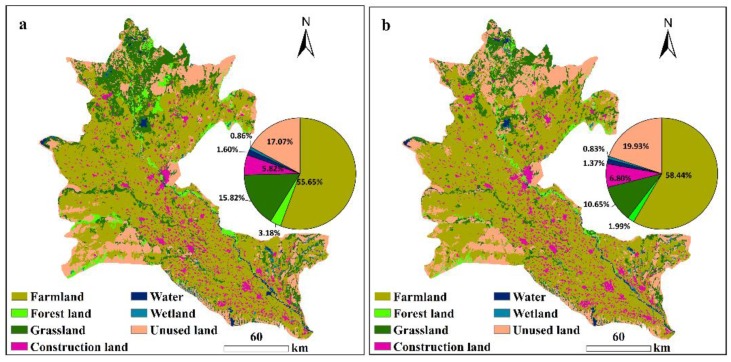
Landscape type maps of the ADD in 2000 (**a**) and 2015 (**b**).

**Figure 4 ijerph-17-00079-f004:**
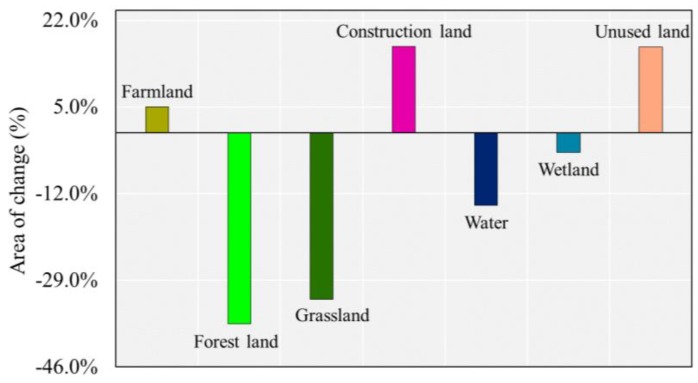
Changes in each landscape during the period from 2000–2015.

**Figure 5 ijerph-17-00079-f005:**
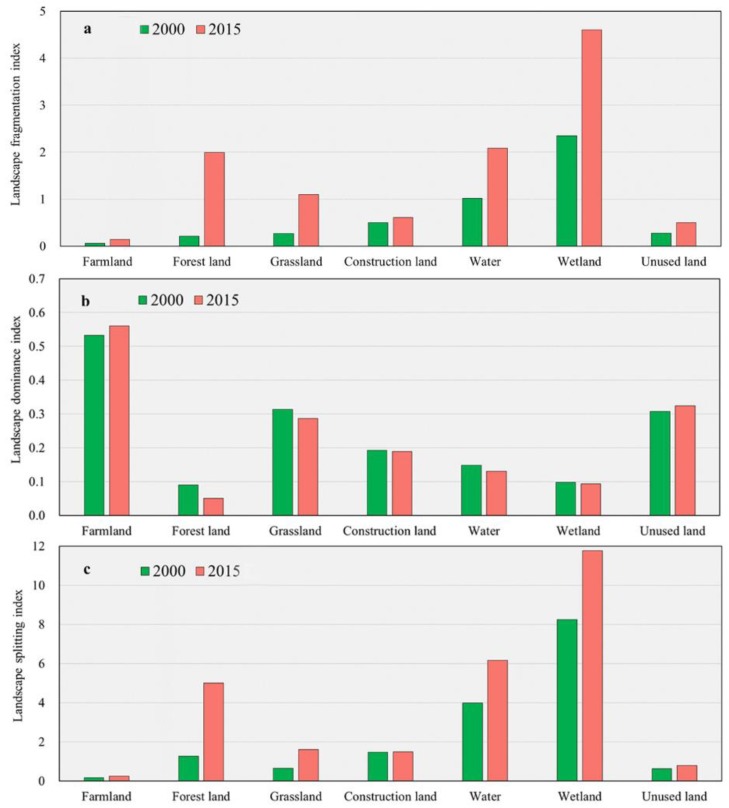
Changes in landscape fragmentation index (**a**), landscape dominance index (**b**) and landscape splitting index (**c**).

**Figure 6 ijerph-17-00079-f006:**
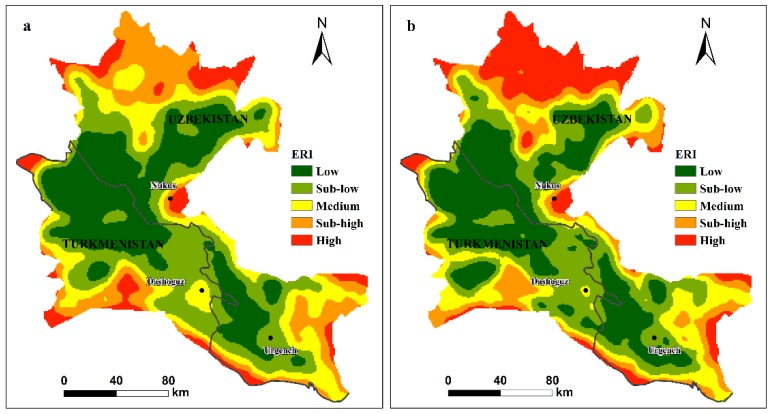
Landscape ecological risk level in the ADD for 2000 (**a**) and 2015 (**b**).

**Figure 7 ijerph-17-00079-f007:**
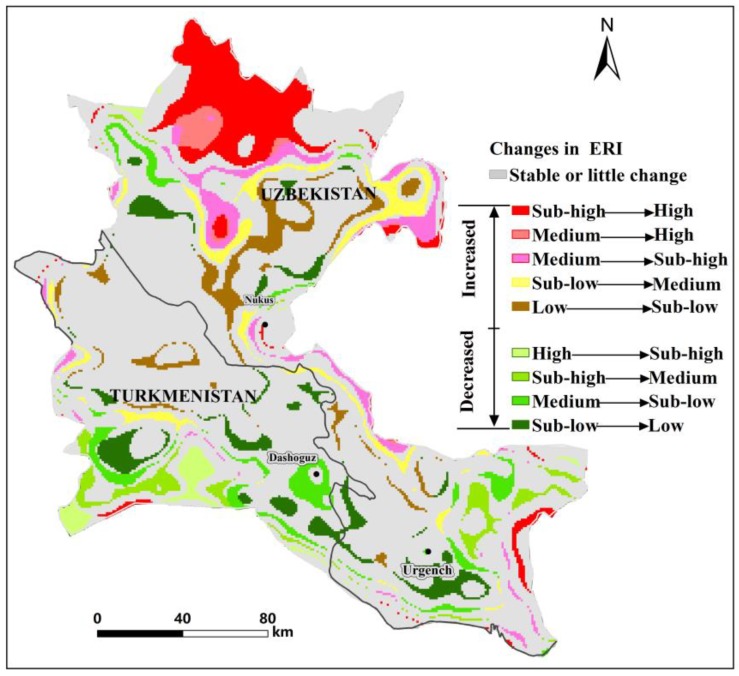
Changes of ecological risk levels between 2000 and 2015.

**Figure 8 ijerph-17-00079-f008:**
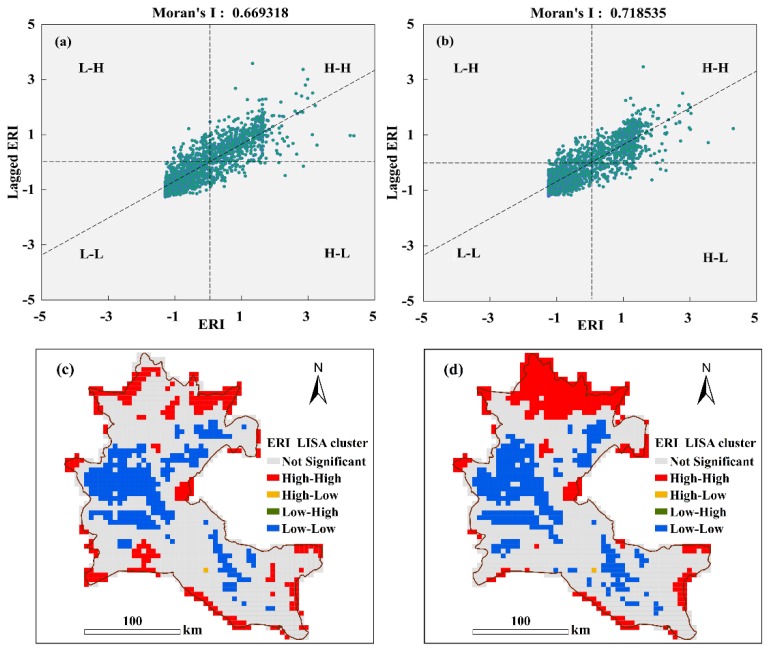
Spatial autocorrelation of the ERI in 2000 (**a**,**c**) and 2015 (**b**,**d**).

**Figure 9 ijerph-17-00079-f009:**
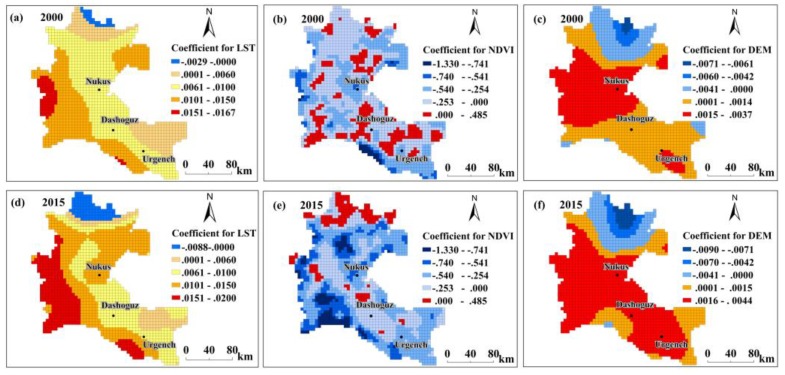
Spatial distribution of the coefficients for the biophysical factors in the geographically weighted regression (GWR) model: (**a**,**d**) land surface temperature (LST); (**b**,**e**) normalized difference vegetation index (NDVI); (**c**,**f**) digital elevation model (DEM).

**Figure 10 ijerph-17-00079-f010:**
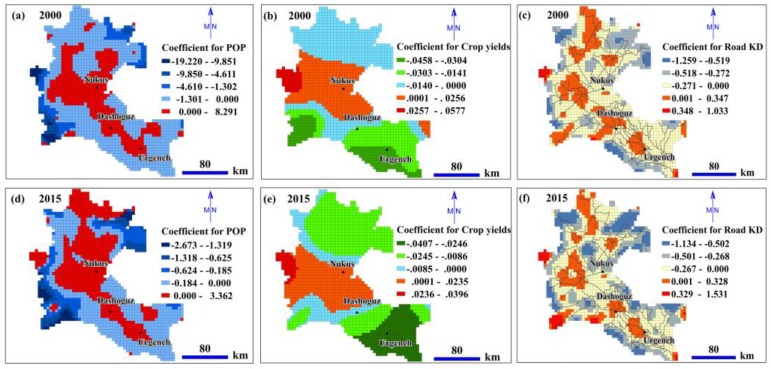
Spatial distribution of the coefficients for the socioeconomic factors in the GWR model: (**a**,**d**) population density (POP); (**b**,**e**) crop yield; and (**c**,**f**) road kernel density (KD).

**Table 1 ijerph-17-00079-t001:** Changes in the proportion of the ecological risk index (ERI) from 2000 to 2015.

Risk Grade	2000		2015		2000–2015	
	Area (km^2^)	Ration	Area (km^2^)	Ration	Area (km^2^)	Ration
Low	10,409.3	28.81%	10,811.8	29.92%	402.5	3.87%
Sub-low	9717.7	26.90%	9231.98	25.55%	−485.72	−5.00%
Medium	6718.05	18.59%	6186.15	17.12%	−531.9	−7.92%
Sub-high	6452.08	17.86%	4625.73	12.80%	−1826.35	−28.31%
High	2832.35	7.84%	5283.65	14.62%	2451.3	86.55%

**Table 2 ijerph-17-00079-t002:** Percentage changes in the ERI from 2000 to 2015 in different regions.

Risk Grade	Low	Sub-Low	Medium	Sub-High	High
UZB (Uzbekistan)	−4.84%	−6.46%	−11.76%	−34.32%	157.57%
TKM (Turkmenistan)	11.93%	−1.19%	−2.99%	−0.08%	−54.83%

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
