# Peer review of "Exploring Variability in Landscape Ecological Risk and Quantifying Its Driving Factors in the Amu Darya Delta"

_ijerph, 2019, doi:10.3390/ijerph17010079_

Round 1

Reviewer 1 Report

Thank you for the opportunity to review your work.  Please see the suggested comments.

Lines 39-47: The first paragraph tries to provide background and problem from various literature, but it needs to be synthesized well.

Lines 49-51: it would be better to provide some examples regarding identified factors.

Lines 82-98: Should the description of case study area move to 2.1 Study area section?

Line 126: Figure 1 is not legible.

Line 130: “A 30 m LULC dataset” needs more explanation. Does it mean 30 meter cell size dataset? Is it raster? But it says shapefile (vector).

Line 150: is “road network data” a part of socioeconomic variables? Is it spatial variable?

Line 318: Figure 7: the figure tries to show all variations, but it is too complex to read the map.

Line 357: You already mentioned GWR is better than OLS at Lines 69-75. Is it necessary to compare these two in here?

Lines 561-563: what are the other factors? These sentences do not have a good connection toward future research.

Author Response

Thank you very much for your valuable comments and kind suggestions on our paper. A point-by-point response to the reviewer’s comments please see the attachment.

Reviewer 2 Report

General comments:

This paper is about the use of a Geographically Weighted regression (GWR) to explore the effects of a series of biogeographic and socioeconomic factors on the Ecological Risk Index (ERI) in the Amu Darya Delta (ADD) area (Uzbekistan and Turkmenistan).

The presented method is interesting, very significative and pretty innovative but it needs to be better explained.

Specific comments:

Introduction

-line 59: "a regional landscape" instead of "an regional landscape"

-line 74: The authors cite two papers about GWR method but they do not cite the first published paper about it. I suggest to improve the references with "Brunsdon, C., Fotheringham, S. and Charlton, M. (1998), Geographically Weighted Regression. Journal of the Royal Statistical Society: Series D (The Statistician), 47: 431-443. doi:10.1111/1467-9884.00145"

Materials and methods

-line 145: Can the authors spend a few words about the Maximum Value Composite (MVC) method? What is it? How does it work?

-line 158: The authors say "roads were extracted in 2015 from OpenstreetMap". Do they have any data for the 2000? Is a comparison between the two studied years possible?

-lines 165-166: Can the authors spend a few words about the mentioned indexes?

-line 187: write "Ecological Risk Index" and the abbreviation in brackets

-line 219: the authors say "a high positive LISA...". What do they mean with high positive? Is there a range of value? Which values can be considered high positive?

-line 222: The authors say "a high negative LISA...". What do they mean with high negative? Is there a range of value? Which values can be considered high negative?

-line 226: write "Geographically Weighted Regression" and the abbreviation in brackets.

-line 232: What do the authors mean with "bandwidth"? Which band?

Results

-lines 261-266: Please try to better explain the percentages. It is possible to make confusion betwwen absolute percentages and relative percentages.

-line 293: write "Ecological Risk Index" and the abbreviation in brackets.

-lines 320-321: The authors are proposing a spatial map of risk from 2000 to 2015. How did they create the map? How many possible combination of factors are there? How many combination do we have in the final map? I suggest to take into consideration the following article "Abandonment of traditional terraced landscape: A change detection approach (a case study in Costa Viola, Calabria, Italy)" doi 10.1002/ldr.2824, in which a similar apporach was proposed as a spatio‐temporal patterns (STePs) map.

-line 339: can the authors spend a few words about GeoDa software? Please improve the text with the download link

-line 356: write "Geographically Weighted Regression" and the abbreviation in brackets.

-lines 378-380: The authors use ERI as indipendent variable and biophysical and socioeconomic variables as explanatory factors. The ERI has been constructed within a grid 5x5km. Dem resolution is 90m. TLS and NDVI resultions is 1km. Did the authors do a resample? Can they explain the process adopted to homogenize heterogeneous data?

Discussion

OK

Conclusion

OK

Author Response

(The authors gave the same response as above.)

Reviewer 3 Report

An interesting paper, which provides a lot of spatio-temporal information about the particular region of the Amu Dayra Delta. It utilizes novel models to assess the ecological risks, which can be useful for research in various contexts. Potentially explaining the impact of human activity on natural ecosystems happening around the globe.

Some minor comments below: 

[line 148] - it would be interesting to mention why the socio-economic data is difficult to obtain. Also is it typical to only use only such variables to describe socio-economic status of a region? Some explanation comes in the discussion section but perhaps it could be at least refereed to initially as well?

[line 557 onwards] - more fine grained approach in explaining the implication for the decision making about the could be added. More precisely, authors talk about "more efficient agricultural production methods", "appropriate ecological policies"and "establishing reliable strategy", but they do not explain what exactly do they mean by these concepts. This ambiguity may lead to misinterpretation of results, while it is very important to correctly translate these results into practice. 

Author Response

(The authors gave the same response as above.)

Round 2

Reviewer 1 Report

Thank you for considering my previous comments and suggestions.

References: Does the revised manuscript include the references from “author_response”? If you use new references to revise/support your paper, you should include those into your manuscript.

Figure 1 is still not legible. It is hard to read the country names from right side map. You might want to remove halo from text.

The revised manuscript should synthesize some of texts/sentences from “author_response”. You have provided some answers from my previous questions, but the revised manuscript too brief to answer my previous questions.

The manuscript needs extensive editing of English language and style. I don’t see any changes compared to previous manuscript.

Reviewer 2 Report

Dear authors,

first of all, thankyou for the the effort in taking into consideration the comments.

Points 1,2,4,5,6,7,8,9,11,12,14 and 15 have been adequately addressed.

Points 3,10,13 and 16 did not adequately addressed.

More in details:
Response 3 - the authors did not provide the explanation of the Maximum Value Composite (MVC) method to a potential reader. In the manuscript there is no explanation about the MVC method and only one reference is cited.

Response 10 - the authors do not have to explain to me the concept of bandwidth in building a GWR process but to a potential reader. Moreover, the references cited in the "Response to reviewer comments" must be correctly reported in the manuscript.

Response 13 - The authors did not response to the questions: How did the authors create the map? How many possible combination of factors are there? How many combination do the authors have in the final map? I think that the authors have to answer, also taking into consideration the suggested paper, also in view of future delelopments of the presented research.

Response 16 - The authors do not explained into manuscript how they did the resampling process (lines276-278) and cited no references.

Round 3

Reviewer 1 Report

Thank you for considering my comments and suggestions. I have no further comments.

Reviewer 2 Report

Dear authors,

thankyou again for the effort in taking into consideration the comments.

All points have been adequately addressed.